# RBS and ABS Coordinated Control Strategy Based on Explicit Model Predictive Control

**DOI:** 10.3390/s24103076

**Published:** 2024-05-12

**Authors:** Liang Chu, Jinwei Li, Zhiqi Guo, Zewei Jiang, Shibo Li, Weiming Du, Yilin Wang, Chong Guo

**Affiliations:** State Key Laboratory of Automotive Simulation and Control, Jilin University, Changchun 130022, China; chuliang@jlu.edu.cn (L.C.); lijw22@mails.jlu.edu.cn (J.L.); zqguo23@mails.jlu.edu.cn (Z.G.); zwjiang22@mails.jlu.edu.cn (Z.J.); lisb22@mails.jlu.edu.cn (S.L.); duwm21@mails.jlu.edu.cn (W.D.); wyl22@mails.jlu.edu.cn (Y.W.)

**Keywords:** four-wheel hub drive electric vehicle, electro-hydraulic composite braking system, coordinated control strategy (CCS), explicit model predictive control (eMPC), error compensator

## Abstract

During the braking process of electric vehicles, both the regenerative braking system (RBS) and anti-lock braking system (ABS) modulate the hydraulic braking force, leading to control conflict that impacts the effectiveness and real-time capability of coordinated control. Aiming to enhance the coordinated control effectiveness of RBS and ABS within the electro-hydraulic composite braking system, this paper proposes a coordinated control strategy based on explicit model predictive control (eMPC-CCS). Initially, a comprehensive braking control framework is established, combining offline adaptive control law generation, online optimized control law application, and state compensation to effectively coordinate braking force through the electro-hydraulic system. During offline processing, eMPC generates a real-time-oriented state feedback control law based on real-world micro trip segments, improving the adaptiveness of the braking strategy across different driving conditions. In the online implementation, the developed three-dimensional eMPC control laws, corresponding to current driving conditions, are invoked, thereby enhancing the potential for real-time braking strategy implementation. Moreover, the state error compensator is integrated into eMPC-CCS, yielding a state gain matrix that optimizes the vehicle braking status and ensures robustness across diverse braking conditions. Lastly, simulation evaluation and hardware-in-the-loop (HIL) testing manifest that the proposed eMPC-CCS effectively coordinates the regenerative and hydraulic braking systems, outperforming other CCSs in terms of braking energy recovery and real-time capability.

## 1. Introduction

Recently, with advancements in vehicle control systems, there has been a steady increase in attention to the coordinated control of electro-hydraulic composite braking systems [1,2]. In electric vehicle braking, the real-time monitoring and processing of data from multiple sensors enable the coordinated control of the regenerative braking system (RBS) and anti-lock braking system (ABS), effectively controlling the distribution of braking torque for each wheel based on parameters like battery status and vehicle speed [3,4,5]. However, the computational burden of data processing and control strategy in the electronic control unit (ECU) of the vehicle can adversely affect its real-time implementation potential. To ensure real-time application capability, robustness, and enhanced braking energy recovery efficiency, developing an advanced coordinated control strategy (CCS) is crucial.

The reported CCSs can be categorized into two groups based on their objectives: optimization-oriented control CCSs [6,7] and real-time capability-oriented CCSs [8,9]. Optimization-oriented control CCSs rely on offline methods to generate reference quantities and trajectories for achieving optimized control performance, such as dynamic programming (DP) [10,11] and particle swarm optimization (PSO) [12,13]. Given prior driving information, optimization-oriented control CCSs can calculate globally optimized control sequences in short-term driving scenarios, thereby maximizing the braking control effect. However, the intensive computational demands of these methods limit their practical application. Additionally, acquiring the necessary driving information is challenging, which restricts the adaptability of these methods across various braking conditions. The real-time capability-oriented CCSs achieve real-time coordinated control by employing various control strategies, including proportional integral derivative-based CCS (PID-CCS) [14,15], linear quadratic regulator-based CCS (LQR-CCS) [16,17], and model predictive control-based CCS (MPC-CCS) [18,19,20]. In PID-CCS, the coordinated control of the RBS and ABS is achieved by tuning the parameters of the PID controller based on the vehicle state and control objectives. However, during the control process, the PID-CCS can only obtain a suboptimal control law, making it difficult to precisely and effectively adjust the coordinated control relationship between the RBS and ABS. Regarding the LQR-CCS, the state gain matrix is formulated, and the wheel braking torque is optimized to attain coordinated control, aligned with the vehicle model and control objectives. Nevertheless, the LQR method requires a linearization process to simplify the control problem and cannot adjust the weights assigned to different time steps within the prediction horizon, alleviating the effect of coordinated braking force distribution. On the contrary, MPC provides flexibility by allowing the weights assigned to different time steps within the prediction horizon to be adjusted, taking into account various variables to make coordinated decisions and obtain the optimal control sequences [21,22]. Additionally, the application of fast MPC techniques, as demonstrated by Chu et al. [23] and Meng et al. [24], further enhances the adaptability and real-time capabilities of MPC in handling dynamic and complex control scenarios. Nonetheless, achieving real-time online solutions to the optimal control problem poses challenges for practical vehicle controllers due to the online rolling optimization process in MPC and the multitude of constraints involved. To overcome this MPC limitation, an alternative approach named explicit model predictive control (eMPC) is proposed [25,26,27]. By introducing multi-parameter quadratic programming (mp-QP), eMPC acquires the explicit solution of state variable and control variable in advance, storing them in the memory inside the controller, thereby transferring online calculation to the offline part to minimize the computational burden. Furthermore, eMPC can be easily compiled on an embedded platform, showing promising real-time application scenarios. Despite its appropriate capability in real-time deployment, eMPC is primarily for generating state feedback control law in linear time-invariant (LTI) systems, which does not allow for optimal control throughout the braking process in highly nonlinear vehicle braking systems [28,29]. Moreover, the initially set constant state variables in the LTI system undergo changes during practical vehicle braking, resulting in state error and diminishing the adaptability and robustness of CCS. Therefore, substantially adjusting the vehicle braking state based on various braking conditions is an intractable task that should be further investigated.

To substantially utilize information collected during vehicle braking, aiming to reduce the adverse impact of state error on the control effect, recent research on optimizing system state variables has introduced adaptive optimization methods [30,31] and state error compensation techniques [32,33,34]. Adaptive optimization methods, such as model reference adaptive control (MRAC) [35], direct adaptive control (DAC) [36], and fuzzy adaptive control (FAC) [37], allow for the automatic adjustment of controller parameters in response to changes in system status and parameters, thereby enhancing control performance. However, the intricate relationship between control parameters and the current state necessitates a comprehensive acknowledgment of the optimized control system or even a burdensome computational modeling process, presenting considerable implementation difficulty. Regarding state error compensation methods, encompassing feedforward error compensating [38,39] and feedback error compensating [40,41], these methods are directed towards diminishing the negative impact of state error generated during the braking process on the control effect through error compensation. For feedforward error compensation, although it can offset system error in advance and effectively suppress external interference, it relies on accurate mathematical models for prediction and faces obstacles in performing nonlinear compensation, posing challenges for implementation in electro-hydraulic composite braking systems. In contrast, feedback error compensation only requires the construction of an error gain matrix from the practical and desired output of the system, facilitating state variable adjustment based on this matrix to bolster system robustness. During the braking process, the vehicle information can be divided into micro trip segments with distinct braking states. Implementing feedback error compensation for each micro trip segment enhances the capability of eMPC-CCS to counteract external interference. To sum up, forming a comprehensive solution is crucial for optimizing the control effectiveness of RBS and ABS while cooperatively ensuring real-time, robust, and adaptive capability. 

In this context, this paper proposes a novel coordinated control strategy based on eMPC, namely the eMPC-CCS, aiming to enhance the real-time capability, adaptability, and robustness of the CCS in the solution. As for the eMPC-CCS, it includes offline control law generation and online control law invocation. In the offline process, a multitude of micro trip segments corresponding to braking operations are collected to generate real-time-oriented state feedback control laws, improving the adaptability of the CCS. During the online implementation, offline-generated state feedback control laws are invoked accordingly to form a 3D eMPC explicit solution in the basic eMPC controller, enhancing the real-time coordinated control of RBS and ABS. Furthermore, a state error compensator is developed to rectify variations in the state variables and integrated into the basic eMPC controller to enhance its functionality. The improved eMPC controller navigates through the 3D eMPC explicit solution using the adjusted state variables and delivers a corrective torque that corrects the braking torque for each wheel, thereby further refining the braking process. Ultimately, simulation evaluation and hardware-in-the-loop (HIL) testing demonstrate the outstanding real-time responsiveness, robustness, and elevated efficiency in the braking energy recuperation of the proposed eMPC-CCS across various braking conditions.

The detailed contributions are illustrated in the following: (1)A coordinated control strategy for the RBS and ABS based on eMPC is proposed, which integrates the offline-generated state feedback control law into online real-time braking to fully enhance the real-time performance of coordinated control.(2)A 3D eMPC law generation method is proposed, which employs the state feedback control law generated at each micro trip segment to formulate an explicit solution for the three-dimensional eMPC, thereby enhancing the adaptability of the control strategy to various braking conditions(3)A state variable optimization method based on feedback error compensation is proposed. This method can integrate the gain matrix into the eMPC-CCS to compensate for the state variable under various braking conditions, improving the ability of the eMPC-CCS to resist external interference.

The remainder of this paper is organized as follows. The general description of the vehicle model’s construction is provided in Section 2. Section 3 elaborates on the developed eMPC-CCS. Section 4 discusses the simulation results. and HIL testing, verifying the superior performance of the raised strategy. The discussions are provided in Section 5. The conclusions are represented in Section 6.

## 2. Modeling

In this paper, a four-wheel hub drive electric vehicle is studied. The corresponding configuration and detailed parameters are, respectively, presented in Figure 1 and Table 1. The electro-hydraulic composite system of the studied vehicle is composed of the motor regenerative braking system and the hydraulic braking system. When the driver presses the brake pedal for deceleration, the vehicle controller calculates the braking intensity and the necessary braking force based on the pedal position. Subsequently, the established CCS guides the motor controller and hydraulic controller in response to the current vehicle states. The motor controller manages the four motors, applying motor braking torque to the wheels, and the hydraulic controller supervises the hydraulic braking unit, facilitating the application of hydraulic braking torque to the respective wheel.

### 2.1. Vehicle Model

The five-degree-of-freedom vehicle dynamics model considering the rotational movement of the four wheels and the longitudinal movement of the vehicle was established in this study.

According to the vehicle dynamics equation, the longitudinal movement can be shown as follows:(1)$$m{\dot{v}}_{x}=-F_{xfl}-F_{xfr}-F_{xrl}-F_{xrr}$$
where m is the vehicle mass, $v_{x}$ is the longitudinal speed, and $F_{xfl}$, $F_{xfr}$, $F_{xrl}$ and $F_{xrr}$ are the ground braking force received by the wheels, respectively.

The rotational motion of the wheels can be written as follows:(2)$$J_{w}{\dot{\omega }}_{ij}=F_{xij}R_{w}-T_{mij}-T_{hij},ij\in \left\{FL,FR,RL,RR\right\}$$
where $J_{w}$ is the wheel moment of inertia, $R_{w}$ is the wheel radius, $T_{mij}$ is the motor braking torque received by the wheels, and $T_{hij}$ is the hydraulic braking torque received by the wheels.

The main parameters of the vehicle model are presented in Table 1.

### 2.2. Tire Model

The structural parameters and mechanical characteristics of tires connecting the vehicle and the road determine the dynamic performance of the vehicle. The magic formula tire model is selected in this study to reflect the dynamic behavior of the tire, which can be written as follows:(3)$$\left\{ \begin{array}{l} F_{x}=D\sin\{C\arctan[B\lambda -E(B\lambda -\arctan(B\lambda ))]\}\;\\ C=a_{0}\\ D=\mu (a_{1}F_{z}^{2}+a_{2}F_{z})\\ B=a_{3}\sin(a_{4}\arctan(a_{5}F_{z}))/(CD)\\ E=a_{6}F_{z}^{2}+a_{7}F_{z}+a_{8} \end{array}\right.$$
where λ is the slip ratio, which can be calculated from $\lambda =(v_{x}-wR_{w})/v_{x}$, μ is the road adhesion coefficient, $F_{x}$ is the longitudinal tire force, and $F_{z}$ is the vertical tire force. In addition, B is the stiffness factor, C is the shape factor, D is the crest factor, E is the curvature factor, and $a_{0}$, ⋯, $a_{8}$ are the model parameters obtained from the experiment.

### 2.3. Motor Model

The purpose of this paper is to investigate the transient response of the motor in the RBS, with the understanding that the dynamic characteristics of the motor will not significantly impact the coordinated control process of the RBS and ABS. The relationship among torque, speed, and efficiency is described by an efficiency map, which is shown in Figure 2. The motor efficiency can be expressed by the following:(4)$$\eta_{em}=\eta (n_{em},T_{em})$$
where $n_{em}$ and $T_{em}$ is the rotating speed and motor torque, respectively.

The mathematical model of the permanent magnet synchronous motor, representing the first-order inertia link, is used to portray the dynamic characteristics during regenerative braking:(5)$$T_{m\_act}=\frac{1}{t_{m}s+1}T_{m\_req}$$
where $T_{m\_act}$ is the actual motor response torque, $T_{m\_req}$ is the target motor response torque, and $t_{m}$ is the motor dynamic response constant.

### 2.4. Battery Model

The effect of temperature on the battery’s operation is ignored, and a simple internal resistance model is used for simulation, motivated by the complexity of temperature effects and the importance of Kirchhoff’s law in establishing the battery voltage balance equation. According to the Kirchhoff law, the battery voltage balance equation can be shown as follows: (6)$$U_{oc}=U+I_{b}R_{\mathrm{int}}$$
where $U_{oc}$ is the open-circuit voltage, U is the load voltage, $I_{b}$ is the battery current, and $R_{\mathrm{int}}$ is the internal resistance of the battery. 

During regenerative braking, the charging current of the battery can be calculated by the following:(7)$$I_{b}(t)=\frac{-U_{OC}(t)+\sqrt{U_{OC}^{2}(t)-4R_{\mathrm{int}}p_{m}}}{2R_{\mathrm{int}}}$$
where $p_{m}$ is the motor power. Additionally, the state of charge (SOC) of the battery is calculated through the ampere-hour integration method:(8)$$\left\{ \begin{array}{l} SOC=SOC_{0}+\frac{Q(t)}{Q_{0}}\\ \dot{Q}(t)=I_{b}(t) \end{array}\right.$$
where $SOC_{0}$ is the initial value of battery SOC, $Q_{0}$ is the battery capacity in Ah, and Q(t) is the variations in battery power.

### 2.5. Hydraulic Braking System Model

The specific working process of the hydraulic braking system, involving the physical properties of the fluid and the response characteristics of hydraulic components, were omitted in this paper as they do not affect electro-hydraulic coordinated control. The response process is considered equivalent to a first-order inertial hysteresis link:(9)$$T_{h\_act}=\frac{e^{-\tau_{h}s}}{t_{h}s+1}T_{h\_req}$$
where $T_{h\_act}$ and $T_{h\_req}$ are the actual braking torque and the target braking torque of the hydraulic system, respectively, $t_{h}$ is the system time constant, and $\tau_{h}$ is the system pure lag time.

## 3. Development of eMPC-CCS

To fully excavate the potential of eMPC-based control solutions in the real-time coordinated control strategy of the RBS and ABS, a comprehensive braking control framework, named the eMPC-CCS, is developed. This approach strengthens real-time capability and adaptability while avoiding control conflict between the RBS and ABS. The implementation process of the eMPC-CCS contains offline control law generation and online control law invocation. An illustration of the eMPC-CCS is shown in Figure 3 and Figure 4.

As depicted in Figure 3, in offline control law generation, vehicle information with diverse braking states, collected from road tests, is segmented into micro trip segments and stored in the vehicle controller. Subsequently, the eMPC algorithm is applied to these distinct micro trip segments, generating the corresponding state feedback control laws, which are then ready for online invocation.

In Figure 4, as for the online implementation, a 3D eMPC explicit solution is employed to invoke the offline control law based on the similarity between the practical braking conditions and micro trip segments. The required braking torque for each wheel under the current braking condition can be allocated according to the ideal braking force distribution strategy, known as the I curve [42]. Ultimately, the eMPC-CCS optimizes the allocation of distributed wheel braking torque by utilizing the matched eMPC explicit solution. It employs the state error compensator and basic eMPC controller within the improved eMPC controller to achieve optimal braking torque allocation within the composite braking system.

### 3.1. Offline Control Law Generation

In the offline processing of eMPC, the state feedback control law of each braking segment of the vehicle in the whole braking process is obtained. The offline control law generation process in eMPC includes several key steps [25]. Firstly, the mathematical model of the system is established. Secondly, the critical region of the system is determined by introducing the theory of mp-QP and the Karush–Kuhn–Tucker (KKT) condition. In each critical region, the control vector can be expressed as an affine function of the state vector, constituting the control law. This affine function remains constant and allows eMPC to derive the control law through offline calculation, thereby eliminating the need for laborious online computation. The calculated optimal control laws are stored in a lookup table for real-time implementation.

#### 3.1.1. Basic eMPC

Before designing the controller, the fundamentals of eMPC are reviewed and the explicit law of the control variable is presented in this section. In eMPC, the solution of an optimization problem includes constraints on the control inputs and system states. The general nonlinear optimization problem in the finite time domain $\left[t_{s},t_{m}\right]$ can be defined as a minimization constraint on the cost function:(10)$$J\left(x,u,p\right)≜\int_{t_{s}}^{t_{m}}L\left(x(t),u(t),p(t_{0})\right)dt+T\left(x(t_{m}),p(t_{0}),t_{m}\right)$$
where x, u, and p are the state vector, control vector, and parameter vector, respectively. In addition, L is the stage cost, and T is the terminal cost. The problem is subject to inequality constraints:(11)$$\;x_{\min}<x(t)<x_{\max}\;$$
(12)$$u_{\min}<u(t)<u_{\max}$$
(13)$$G\left(x(t),u(t),p(t_{s}),t\right)\leq 0$$

The ordinary differential equations (ODEs) describing the system dynamics represent the equality constraints:(14)$$\frac{d}{dt}x(t)=f\left(x(t),u(t),p(t_{s}),t\right)$$

In the process of formulating the control problem, the prediction horizon is defined as tq=Nptr, where Np is the number of prediction steps and tr is the discretization interval of the internal model. It is assumed that the control input u[ts,tm] is calculated by u(t)=ξ(U,t), where U is the control parameter vector. The optimal control problem is now in its mp-NLP generic form:(15)J*(x(ts),p(ts))=minUJ(x(ts),U,p(ts))s.t.G(x(ts),U,p(ts))≤0
where p includes the system and controller parameters, which are considered constant for the duration of the prediction horizon. One additional vector is defined as xp(ts)∈ℝnp, where np=n+d, i.e., np is the sum of the number of state n and the size of parameter d:(16)xp(ts)=[x(ts),p(ts)]

Hence, based on Equation (16), the cost function can be reformulated as follows:(17)$$\begin{array}{c} J^{*}\left(x_{p}(t_{s})\right)=\underset{U^{*}}{\min}J\left(x_{p}(t_{s}),U\right)\\ s.t.G\left(x_{p}(t_{s}),U\right)\leq 0 \end{array}$$

In this study, the mp-QP problem is adopted to solve the mp-NLP problem by its approximation [43,44]. The mp-NLP in Equation (17) is linearized around a predefined point $\left(x_{p,0},U_{0}\right)$ by utilizing Taylor expansion. Therefore, the cost function is approximated with a quadratic function and the constraints assume a linear formulation.
(18)$$\begin{array}{c} J_{0}(x_{p},U)≜\frac{1}{2}{\left(U-U_{0}\right)}^{T}H_{0}(U-U_{0})+\left(C_{0}+{\left(x_{p}-x_{p,0}\right)}^{T}F_{0}\right)(U-U_{0})+Y_{0}(x_{p})\\ s.t.G_{0}(U-U_{0})\leq W_{0}+S_{0}(x_{p}-x_{p,0}) \end{array}$$

By evaluating at the linearization point $\left(x_{p,0},U_{0}\right)$, the different terms in Equation (18) can be computed as follows:(19)$$H_{0}≜\nabla_{UU}^{2}J(x_{p,0},U_{0})C_{0}≜{\left(\nabla_{U}J(x_{p,0},U_{0})\right)}^{T}$$
(20)$$F_{0}≜\frac{1}{2}\left({\left(\nabla_{x_{p}U}^{2}J(x_{p,0},U_{0})\right)}^{T}+{\left(\nabla_{Ux_{p}}^{2}J(x_{p,0},U_{0})\right)}^{T}\right)$$
(21)$$\begin{array}{r} Y_{0}(x_{p})≜\frac{1}{2}{\left(x_{p}-x_{p,0}\right)}^{T}\nabla_{x_{p}x_{p}}^{2}J(x_{p,0},U_{0})(x_{p}-x_{p,0})\\ +{\left(\nabla_{x_{p}}J(x_{p,0},U_{0})\right)}^{T}(x_{p}-x_{p,0})+J(x_{p,0},U_{0}) \end{array}$$
(22)$$G_{0}≜{\left(\nabla_{U}G(x_{p,0},U_{0})\right)}^{T}$$
(23)$$W_{0}≜-G(x_{p,0},U_{0})$$
(24)$$S_{0}≜-{\left(\nabla_{x_{p}}G(x_{p,0},U_{0})\right)}^{T}$$

The mp-QP problem is employed to compute local approximations of the original mp-NLP problem in the exploration space. This is represented as multiple hyperrectangles, on which single mp-QP problems are solved. Each hyperrectangle is further partitioned into polyhedra, i.e., the CRregions of the mp-QP problem. Finally, the mp-QP solution is represented as a piecewise affine (PWA) function, which is determined by a finite number of regions dividing state variables. Given that eMPC is designed for LTI systems, achieving improved control effects in the coordinated control of the RBS and ABS requires both offline and online optimizations.

#### 3.1.2. Internal Prediction Model

In the basic design of the eMPC controller, only the control process of the left front (FL) brake component is depicted, with the other wheels following a similar pattern.

In the internal quarter-vehicle model, the slip ratio of the left front wheel, $\lambda_{FL}$, is defined as follows:(25)$$\lambda_{FL}(t)=\frac{v_{x}(t)-\omega_{FL}(t)R_{w}}{v_{x}(t)}$$
where $v_{x}$ is the longitudinal speed, $\omega_{FL}$ is the wheel velocity, and $R_{w}$ is the wheel radius. The time derivative of $\lambda_{FL}$ can be expressed as follows:(26)$$\frac{d}{dt}\lambda_{FL}=\frac{\frac{d}{dt}v_{x}(t)-R_{w}\frac{d}{dt}\omega_{FL}(t)}{v_{x}(t)}-\frac{\left(v_{x}(t)-\omega_{FL}(t)R_{w}\right)\left(\frac{d}{dt}v_{x}(t)\right)}{v_{x}^{2}(t)}$$
where the wheel rotation dynamics equation can be expressed as follows:(27)$$\frac{d}{dt}\omega_{FL}(t)=\frac{1}{J_{w}}(F_{x,FL}R_{w}-T_{CA,FL}+\Delta T_{FL}(t))\;$$
where $J_{w}$ is the wheel moment of inertia, and $T_{CA,FL}$ is the distributed braking torque according to the I-curve, which is kept constant over the prediction horizon. The longitudinal force balance of the quarter-vehicle model associated with the wheel under consideration can be shown as follows:(28)$$\frac{d}{dt}v_{x}(t)=-\frac{1}{m_{F}}F_{x,FL}$$
where $F_{x}$ is the longitudinal tire force, which can be calculated using the simplified Pacejka magic formula (MF) model [45], as follows:(29)$$\left\{ \begin{array}{l} F_{x,FL}=\mu_{x,FL}F_{z,F}\\ \mu_{FL}=D\sin\{C\arctan[B\lambda_{FL}-E(B\lambda_{FL}-\arctan(B\lambda_{FL}))]\} \end{array}\right.$$
where $F_{z,F}$ is the vertical tire load, considered a constant, and $\mu_{x,FL}$ is the longitudinal tire–road friction coefficient. Additionally, B, C, D and E are the MF parameters, which are shown in Table 2. 

Therefore, the time derivative of Equation (25) can be rewritten as follows:(30)$$\frac{d}{dt}\lambda_{FL}(t)=\frac{\left(\lambda_{FL}(t)-1\right)}{v_{x}(t)}\frac{F_{x,FL}}{m_{F}}-\frac{R_{w}}{J_{w}v_{x}(t)}\left(F_{x,FL}(t)R_{w}-T_{CA,FL}(t)+\Delta T_{FL}(t)\right)$$

Due to the difference in inertia, the longitudinal dynamic of the vehicle is much slower than the rotational dynamic of the wheels. Therefore, the vehicle speed is considered a slowly changing parameter.

An integral process is incorporated into the prediction model to tackle the steady-state error and model uncertainty. Hence, the model includes $e_{FL}(t)$, which is the integral of the error between the actual slip rate, $\lambda_{FL}$, and the reference slip rate, ${\lambda_{FL}}^{ref}$.
(31)$$\frac{d}{dt}e_{FL}(t)=\lambda_{FL}(t)-{\lambda_{FL}}^{ref}$$

For each sampling moment, the state equation of the system after linearization can be obtained as follows:(32)x˙(t)=A(t)x(t)+B(t)u(t)
where x(t)∈ℝn, u(t)∈ℝm
(33)A(t)=[Fx,FL(t)vx(t)⋅mF+(λFL(t)−1)vx(t)⋅mF⋅dFx,FL(t)dλFL(t)−Rw2Jw⋅vx(t)⋅dFx,FL(t)dλFL(t)010]
(34) B(t)=[−RwJw⋅vx(t)0]
where dFx,FL(t)dλFL(t) denotes the derivative of Fx,FL(t) to λFL(t).

The continuous-time state space representation, Equation (32), is discretized with a sampling time.
(35)$$x(k+1)=A_{d}[k]x(k)+B_{d}[k]u(k)$$
where
(36)$$A_{d}[k]=\left[\begin{array}{cc} A_{d}^{1}[k] & 0\\ t_{s} & 1 \end{array}\right]$$
(37)$${\;B}_{d}[k]=\left[\begin{array}{c} A_{d}^{2}[k]\\ 0 \end{array}\right]$$
where $A_{d}^{i}[k]$, i=1,2 indicates the varying elements due to $v_{x}(t)$, and k is the sampling time index.

In the prediction model, the state vector, input vector, and parameter vector are $x_{FL}=[\lambda_{FL},e_{FL}]$, $u=[\Delta T_{FL}]$, and $p=\left[v_{x},T_{CA,FL},{\lambda_{FL}}^{ref}\right]$, respectively. The prediction horizon, $t_{p}=N_{p}t_{r}$ (i.e., $N_{p}$ is prediction steps; $t_{r}$ is sampling time), is selected for the current implementation. The problem includes five parameters (a 5D-eMPC problem), i.e., $x_{p,FL}=\left[\lambda_{FL}(t_{s}),e_{FL}(t_{s}),v_{x}(t_{s}),T_{CA,FL}(t_{s}),{\lambda_{FL}}^{ref}\right]$, and four decision variables, i.e., $U_{FL}=\left[\Delta T_{FL}(t_{s}),\Delta T_{FL}(t_{s+1}),\Delta T_{FL}(t_{s+2}),\Delta T_{FL}(t_{s+3})\right]$. The horizon control input that is applied to the system is $u=\Delta T(t_{s})$, which will be indicated as u in the remainder. The 5D-eMPC problem will be referred to as eMPC_5D_ in the remainder. 

At each moment throughout the vehicle braking process, the state variables at that moment are aligned with the previously gathered braking segments, facilitating the generation of the state feedback control laws depicted in Figure 5 within the basic eMPC controller. Subsequently, the generated control laws are prepared for online invocation, reducing the computational burden of the control problem.

#### 3.1.3. Cost Function and Constraints

During the offline optimization process, the optimal adjustment of the required braking torque for the FL wheel is achieved by minimizing the cost function. The changes in braking torque are applied to the wheels to quickly achieve the desired slip rate of the FL wheel throughout the ABS braking process. Moreover, the cost function includes the influence of braking torque on the wheel to minimize variations in wheel braking torque. The cost function can be written as follows:(38)$$J_{FL}=\int_{t_{k}}^{t_{f}}\left[\frac{q_{1}}{w_{1}^{2}}{\left(\lambda_{FL}(t)-{\lambda_{FL}}^{ref}\right)}^{2}+\frac{q_{2}}{w_{2}^{2}}e_{FL}{\left(t\right)}^{2}+\frac{r_{u}}{w_{u}^{2}}\Delta T_{FL}{\left(t\right)}^{2}\right]dt+\frac{p_{1}}{w_{1}^{2}}{\left(\lambda_{FL}(t_{f})-\lambda^{ref}\right)}^{2}+\frac{p_{2}}{w_{2}^{2}}e_{FL}{\left(t_{f}\right)}^{2}$$
where $q_{1}=5$, $q_{2}=60$, $r_{u}=10$, $p_{1}=5$, $p_{2}=60$, $w_{1}=0.1$, $w_{2}=0.1$, and $w_{u}=3000$. In addition, ${\lambda_{FL}}^{ref}$ is computed from the longitudinal tire force characteristics as a function of the slip ratio, by using the MF [46].

For the implementation of eMPC, some constraints related to the powertrain performance and vehicle dynamic should be set, which would ensure that different components can operate within limits. Thus, the inequality constraints of Equation (38) can be shown as follows:(39)$$\lambda_{\min}<\lambda_{FL}(t)<\lambda_{\max}$$
(40)$$\Delta T_{\min}<\Delta T_{FL}(t)<\Delta T_{\max}$$
where $\Delta T_{\max}=T_{CA,FL}$ and $\Delta T_{\min}=0$, while $\lambda_{\min}$ and $\lambda_{\max}$ are used as tuning parameters.

### 3.2. Online Implementation

Building upon the control laws generated in the preceding offline process, the online implementation invokes the eMPC solution generated by the three-dimensional eMPC control law generation method for the adaptability of the CCS through various braking conditions. To improve the ability of the eMPC-CCS to resist external interference, a state error compensator is developed, optimizing the control logic of the control problem. The illustration of the online implementation of the eMPC-CCS is shown in Figure 6. During the control process, the state error compensator is used to optimize the state variables, and then the optimized state variables are transmitted to the 3D eMPC solution in the basic eMPC controller to obtain a correction torque to correct the braking torque of each wheel. The corrected braking torque prioritizes the requirements of the motor before being distributed to both the motor and the hydraulic braking system, thus ensuring the enhanced adaptability and robustness of the CCSs.

#### 3.2.1. Explicit Solution of RBS and ABS Coordinated Control

Regarding the comment about eMPC_5D_, the representation of the explicit solution is presented in Figure 7. Figure 7 illustrates that the three-dimensional eMPC control law generation method partitions the state input space into polyhedral regions, each connected via PWA systems. The principle of PWA in eMPC involves segmenting the state space of the system and using linear models and corresponding optimal control laws within each region to effectively control nonlinear systems. The offline control law generation stage precomputes and stores optimal control laws, while the online control law invocation stage dynamically selects and applies the appropriate control law based on the current system state, achieving real-time control of the system.

Invoking these offline generated control laws across diverse state input domains facilitates optimal control input determination, amalgamating into an eMPC explicit solution. To visualize the three-dimensional eMPC explicit solution in Figure 7, three parameters are treated as constants: the integral of the slip rate error, $x_{p,FL}(2)$, is 0, the vehicle speed, $x_{p,FL}(3)$, is 100 km/h, and the reference slip ratio, $x_{p,FL}(5)$, is set to 0.15. Furthermore, $x_{p,FL}(4)$ is the demand torque calculated using the I-curve and the red dashed line is indicated as a constant slip ratio reference in the legend.

Therefore, the solution consists of three planes, including the following:(1)A plateau of zero-control input for the low slip ratio indicates an input lower constraint in Figure 7. According to Equations (39) and (40), the torque correction must be positive.(2)An inclined plane, parallel to the $x_{p,FL}(1)$ axis, indicates an input upper constraint in Figure 7. According to Equations (39) and (40), the regulating torque cannot be larger than the demand torque.(3)Another inclined plane is the non-saturated feedback control input.

In the state prediction process of eMPC, the precision of the state variable markedly impacts the efficacy of vehicle control. Due to the eMPC solution being obtained through linearization, it encounters limitations in the mathematical state function that represents the change in braking torque, making it challenging to fully reflect the dynamic characteristics of the electro-hydraulic composite braking system. Therefore, a state error compensator that can address state error within the predictive horizon is crucial for enhancing the effectiveness of the eMPC controller, which will be elaborated upon in the subsequent section.

#### 3.2.2. Online State Optimization Based on Feedback Error Compensation

To precisely anticipate changes in braking torque within the predictive horizon, an algorithm [47] is proposed to compensate for the state variable x(t) in Equation (32). This algorithm enhances the robustness of the eMPC controller against parameter changes.

Due to the vehicle speed being fixed during a sampling time, the matrices $A_{d}[k]$ and $B_{d}[k]$ in Equation (35) are regarded as the constant matrices $A_{d}$ and $B_{d}$, and the system is transformed into an LTI system. The corresponding state space representation can be shown as follows: (41)$$x(k+1)=A_{d}x(k)+B_{d}u(k)$$

However, maintaining a constant longitudinal velocity, $v_{x}$, in an actual driving situation is challenging. In this paper, $v_{x}$ is regarded as a slowly varying parameter, $v_{x}(t)$. 

The time-varying matrix Ad[k] can be determined as follows:(42)Ad[k]=Ad+∑i=1NvΔx,iαx,i(k)
where Δx,i∈ℝn×n, αi(t)∈ℝ, and Nv represent the number of the time-varying elements in the matrix Ad[k].

In this study, $A_{d}[k]$ is assumed to be a time-varying matrix with sufficiently slow variation, but the mechanisms of its evolution are not clear. Additionally, the effect of the time-varying matrix $B_{d}[k]$ is neglected, and it is treated as a constant matrix. Thus, $A_{d}$, $B_{d}$ and the matrix $\Delta_{x,i}$ are known. However, the function $\alpha_{x,i}(k)$, describing the time-varying part of $A_{d}[k]$, remains unclear.

Equation (35) can be rewritten as follows:(43)$$x(k)-A_{d}x(k-1)=\sum_{i=1}^{N_{v}}\left[\Delta_{x,i}\alpha_{x,i}(k-1)x(k-1)\right]$$

Assuming that Dx(k−1)=[Δx,1x(k−1),…,Δx,Nvx(k−1)], Δx,ix(t−1)(1≤i≤Nv) is a column vector in ℝn, Equation (43) can be rewritten as follows:(44)x(k)−Adx(k−1)=Dx(k−1)[αx,1(k−1),…,αNv(k−1)]

In Equation (44), the previous states, x(k−1), and input, u(k−1), are known via measurement, so the current state, x(k), can be calculated. To simplify the study in this section, the ranks of $D_{x}(k-1)$ and $\left[D_{x}(k-1),x(k)-A_{d}x(k-1)-B_{d}x(k-1)\right]$ are considered to be equal for all k≥1. Therefore, based on the Rouché–Capelli theorem [48], Equation (44) has a unique solution, $\alpha_{x,i}(k-1)(1\leq i\leq N_{v})$, for all k≥1. 

As a result, $A_{d}[k-1]$ can be obtained from the most recent state, x(k−1), control history, u(k−1), and current state, x(k). To obtain the state gain matrix, L(k), for compensating the state vector, the equation is expressed as follows:(45)$$\hat{x}(k+1)=A_{d}\hat{x}(k)+B_{d}u(k)$$
where $\hat{x}(k)$ denotes a compensated state vector at the time instant k. $A_{d}$ is considered to be invertible, and $\hat{x}(k)$ in Equation (45) is replaced by L(k)x(k), where L(k) can be shown as follows: (46)$$L(k)=A_{d}^{-1}A_{d}(k)$$

Then, $\hat{x}(k+1)=x(k+1)$ in Equations (35) and (45), meaning the compensated state vector can be identical to the state vector obtained by the time-varying system matrix, $A_{d}[k]$, in Equation (35). By substituting $A_{d}[k-1]$ from Equation (44) for $A_{d}[k]$ in Equation (46), L(k−1) can be calculated. With sufficiently slow time-varying parameters, it can be assumed that L(k)≅L(k−1). Consequently, the compensated state vector can be shown as follows:(47)$$\hat{x}(k)=L(k)x(k)≅L(k-1)x(k)$$

The online implementation structure of the state error compensation is represented in Table 3.

Moreover, considering the research target in the study is to ensure braking stability while improving the braking energy recovery rate of electric vehicles, the corresponding system parameter vector, $x_{p,FL}$, is rewritten as follows:(48)$${\hat{x}}_{p,FL}=\left[{\hat{\lambda }}_{FL}(t_{k}),e_{FL}(t_{k}),v_{x}(t_{k}),T_{CA,FL}(t_{k})\right]$$
where ${\hat{\lambda }}_{FL}(t_{k})={\hat{\lambda }}_{FL}(t_{k})+A_{d}^{-1}A_{d}(t_{k})$, and other system parameters are the same as in $x_{p,FL}$.

The architecture of the improved eMPC controller, which consists of the basic eMPC controller and the state error compensator, is shown in Figure 8. The basic eMPC controller can generate critical regions in Figure 8 when only $x_{p,FL}(1)$ and $x_{p,FL}(4)$ vary and the other parameters are fixed. Each region owns its unique sequence of optimal vectors, and the controller explicitly chooses a region based on the parameter values. The state error compensator employs feedback error compensation according to the variation in the system matrix, i.e., ${\hat{x}}_{p,FL}$ in Equation (48), and enhances the robustness of the controller against parameter variation.

Equation (48) dictates that compensating the state variable in Figure 8 involves straightforward matrix multiplication considering the previous state and input. This compensator avoids the necessity for modifying the critical regions of eMPC concerning parameter variation.

In summary, a comprehensive braking control framework is established by combining offline adaptive control law generation, online optimal control law application, and state compensation.

## 4. Comparison of Simulation Results

In this section, based on the studied vehicle and the proposed structure of the electro-hydraulic composite braking system, a MATLAB/Simulink simulation platform is established for a four-wheel hub drive electric vehicle. The coordinated control of the electro-hydraulic composite braking system emphasizes regulating the electric and hydraulic braking force to ensure braking safety in both regular and emergency braking conditions. Additionally, the regenerative braking system should provide a certain amount of electric braking force during the test, and the test conditions need to include regular braking and emergency braking conditions. Therefore, the performance of the proposed eMPC-CCS is evaluated in comparison with that of conventional CCSs, such as the PID-CCS, LQR-CCS, and MPC-CCS. Evaluations include assessing the performance of the driver in an integrated braking operation, starting with low-intensity braking and then transitioning to high-intensity braking, to verify the effectiveness of the proposed eMPC-CCS. Another assessment involves emergency braking on joint and bisectional roads to verify the braking stability and robustness of the proposed eMPC-CCS. The initial battery SOC is set to 0.8. The key parameters of the vehicle model are shown in Table 1 and Table 2. According to the above analysis, the simulated braking conditions selected in this section are as follows:The vehicle performs integrated braking on the road with a high-adhesion coefficient of 0.8 and the initial braking speed is 100 km/h;The vehicle performs integrated braking on the road with a low-adhesion coefficient of 0.3 and the initial braking speed is 50 km/h;The vehicle performs emergency braking on the joint road with an adhesion coefficient that changes from 0.2 to 0.8, and the initial braking speed is 60 km/h;The vehicle performs emergency braking on the bisectional road with an adhesion coefficient of 0.3 on the left and 0.8 on the right, and the initial braking speed is 60 km/h.

### 4.1. Simulation Analysis of Integrated Braking Conditions

As shown in Figure 9 and Figure 10, the integrated braking conditions are simulated and analyzed for vehicle braking on the high- and low-adhesion coefficient road, respectively. In addition, the driver is set to perform an integrated braking operation, i.e., small-intensity braking with a value of 0.1 at 0–1 s first, which changes to large-intensity braking with a value of 1 after 1 s. When the vehicle brakes on this road, the front-left wheel is selected to be analyzed, i.e., the simulation results of the four coordinated control strategies controlling whole braking are demonstrated by the front-left wheel motion. 

The vehicle braking simulation results on the dry asphalt road with an adhesion coefficient of 0.8 are shown in Figure 9. The initial braking speed is 100 km/h and the battery SOC is 0.8. In Figure 9d–f, within 1 s from braking, according to the current vehicle status, the braking mode analysis module judges the braking system working mode to be conventional regenerative braking. At this time, the regenerative braking system can completely provide the whole vehicle’s demanded braking torque. After 1 s, the braking intensity is more than 0.2, and the eMPC-CCS distributes the front and rear axle braking force according to the I-curve and prioritizes the motor to provide the demanded braking force. When the demanded braking force is larger than the maximum braking force that can be provided by the motor, the motor outputs the current maximum braking force, and the remaining demanded braking force is provided by the hydraulic braking system. In Figure 9a–c, with the increase in braking force applied to the wheels, the braking system enters into the RBS and ABS coordinated control mode. Then, the RBS and ABS coordinated control strategy is triggered to control the wheels for antilock control, at which point the wheels engage in emergency braking. When the wheel reaches the ABS-triggering condition, the eMPC-CCS calculates the wheel demand braking torque according to the vehicle state parameters, to maintain the slip ratio at the optimal road slip ratio. It can be seen from Figure 9a that during the emergency braking, compared with the other three control strategies, the eMPC-CCS can quickly stabilize the slip rate around 0.15 with minimal fluctuation. This means that the adhesion between the wheels and the road is more stable, and makes the brake system control the wheels more accurately. When the vehicle speed drops to 10 km/h, the braking torque of the motor decreases to 0 Nm and the motor exits the braking process. After that, the vehicle is not suitable for anti-lock braking, so the coordinated control strategy controls the hydraulic braking torque to increase rapidly to complete the final parking brake. 

The vehicle braking simulation results on a snowy road with an adhesion coefficient of 0.3 are shown in Figure 10. The initial braking speed is 50 km/h, and the battery SOC is 0.8. In Figure 10d–f, within 1 s from braking, according to the current vehicle status, the braking mode analysis module judges the braking system working mode to be conventional braking. At this time, the regenerative braking system can completely provide the demanded braking torque. After 1 s, the braking intensity is more than 0.2, and the eMPC-CCS distributes the front and rear axle braking force according to the I-curve and prioritizes the motor to provide the demanded braking force. At this point, large fluctuations in the slip ratio occur due to emergency braking taking place, and the four coordinated control strategies perform anti-lock braking by adjusting the braking torque on the wheels. In Figure 10a, compared with the other three control strategies, the eMPC-CCS can maintain the slip rate around 0.15 faster and more stably with minimal fluctuation. When the vehicle speed is less than 10 km/h, the wheel speed drops directly to 0, the regenerative braking system quits braking, and the required braking force is completely provided by the hydraulic braking system, until complete parking.

By comparing the braking performance indicators among the four coordinated control strategies, shown in Figure 11, the eMPC-CCS has the maximum energy recovery efficiency and adhesion factor utilization rate on high- and low-adhesion-coefficient roads. The MPC-CCS performs a bit worse than the eMPC-CCS but better than the LQR-CCS and PID-CCS; the PID-CCS has the worst performance. The differences in performance among these strategies are due to the optimal control logic invocation and the method of achieving the invoking process. The improved eMPC controller can reasonably distribute the braking force of each wheel and solidly underpin the control logic update, ensuring the motor is in a high-efficiency field in different driving conditions. The MPC-CCS with driving condition identification in the whole braking process can also invoke the optimized control logic but struggles to obtain a flexible, timely update in contrast to the eMPC-CCS. 

Table 4 and Table 5 present the simulation results of various CCSs, emphasizing energy recovery efficiency as the main evaluation criterion for assessing the coordinated control effectiveness of the RBS and ABS. As shown in Equation (49), the vehicle speed changes from $v_{0}$ to $v_{1}$ during the time interval $\left[0,t_{b}\right]$.
(49)$$\eta =\frac{\int_{0}^{t_{0}}\left(1000(\sum_{i=1}^{4}T_{i}\cdot \omega_{i}\cdot \eta_{i}(T_{i},\omega_{i}))/9550\right)dt}{m(v_{0}^{2}-v_{1}^{2})/2}$$
where $T_{i}$ and $\omega_{i}$ are the braking torque and speed output by the motors, respectively. $\eta_{i}(i=1,2,3,4)$ is the efficiency of motors when outputting torque, $T_{i}$, and speed, $\omega_{i}$. m is the vehicle mass, and η is the braking energy recovery efficiency.

According to the numerical results in Figure 11a and Table 4, the eMPC-CCS contributes to an increase of 8.02% in the energy recovery efficiency compared with the MPC-CCS, an increase of 9.39% compared with the LQR-CCS, and an increase of 11.75% compared with the MPC-CCS on a high-adhesion-coefficient road. Similarly, numerical results in Figure 11b and Table 5 reveal that the eMPC-CCS contributes to an increase of 4.05% in the energy recovery efficiency compared with the MPC-CCS, an increase of 17.62% compared with the LQR-CCS, and an increase of 12.43% compared with the MPC-CCS on a low-adhesion-coefficient road. From the perspective of energy recovery efficiency, the eMPC-CCS can more effectively invoke the optimized control thresholds in real time by referring to the wheel braking torque corresponding to the optimal road slip ratio, so it has better adaptability to different braking modes. In terms of the adhesion factor utilization rate in Table 4 and Table 5, the eMPC-CCS can improve by 1.4% and 1.2% compared with the MPC-CCS, which means that the vehicle has more braking force on the road, can transfer power more efficiently, and provides better handling performance and stability.

### 4.2. Robustness Verification for eMPC-CCS

As shown in Figure 12 and Figure 13, braking simulations are performed on the joint road with varying adhesion coefficients, and a bisectional road with different adhesion coefficients on the left and right sides, respectively.

The vehicle braking simulation results of four coordinated control strategies on the joint road are shown in Figure 12. At the initial braking stage, emergency braking on a snowy road with an adhesion coefficient of 0.3 is simulated; the initial braking speed is 60 km/h and the initial SOC is 0.8. In Figure 12d–i, With the increase in braking torque applied to the wheels, the front and rear wheels reach the condition of triggering the ABS one after another. After the ABS is triggered, the proposed eMPC-CSS calculates the demanded braking torque for the wheels to maintain the optimal slip rate, and then controls the braking system to apply the corresponding braking torque on the wheels to quickly stabilize the slip rate near the optimal slip rate. After the vehicle has traveled 15 m, i.e., after 1 s, the road suddenly becomes a dry asphalt road with a value of 0.8. The proposed eMPC-CSS can still quickly make the wheel slip ratio track the best slip ratio on the current road. In Figure 12b,c, it is clear that the slip rate under the eMPC-CSS fluctuates more gently than that under the other three control strategies, and can quickly keep up with the desired slip rate and stay close to it so that the wheels do not lock up during ABS braking. The smaller fluctuations in the slip rate allow the braking system to control the wheels more accurately and can improve vehicle stability and reduce unnecessary vehicle sway and drift due to braking. Thus, the eMPC-CSS has better stability and robustness on the joint road.

As shown in Figure 13, the vehicle performs emergency braking on the bisectional road with an adhesion coefficient on the left side of 0.3 and an adhesion coefficient on the right side of 0.8. The initial braking speed is 60 km/h and the initial SOC is 0.8. At the beginning of braking, as the braking torque increases, the front-left wheel on the low-adhesion-coefficient road triggers the ABS earlier than the front-right wheel on the high-adhesion-coefficient road. At this time, there is a large gap between the slip rate of the front-left wheel and the front-right wheel. The eMPC-CSS controls the slip rate of the front-left wheel to be near the optimal road slip rate while continuing to increase the braking torque on the front-right wheel. Subsequently, the rear-left wheel, which is on the low-adhesion-coefficient road, reaches the ABS-triggering condition, and the eMPC-CSS controls the slip rate of the rear-left wheel near the optimal road slip rate while continuing to increase the braking torque on the rear-right wheel. Finally, the front-right wheel, which is on a high-adhesion-coefficient road, also reaches the ABS-triggering condition. Aiming at providing good braking performance, the eMPC-CSS controls the wheel slip rate near the optimal road slip rate, which utilizes the road adhesion conditions to the fullest. When the vehicle speed is more than 10 km/h, the left and right wheels are not locked to ensure stability and safety during braking.

As shown in Figure 14, compared with the braking performance indicators among the four coordinated control strategies, the eMPC-CCS has the maximum energy recovery efficiency and braking deceleration on the joint road and the bisectional road. In addition, four coordinated control strategies can stabilize the vehicle during braking, but the eMPC-CCS has the shortest braking distance. The MPC-CCS performs a bit worse than the eMPC-CCS but better than the LQR-CCS and PID-CCS; LQR-CCS has the worst performance. The eMPC-CCS can reasonably distribute the braking force of each wheel, and make corrections to ensure that the motor is in the field of high efficiency under different driving conditions. 

According to the numerical results in Figure 14a and Table 6, the eMPC-CCS contributes to an increase of 5.63% in the energy recovery efficiency compared with the MPC-CCS, an increase of 12.86% compared with the LQR-CCS, and an increase of 8.33% compared with the MPC-CCS on the joint road. Similarly, numerical results in Figure 14b and Table 7 reveal that the eMPC-CCS contributes to an increase of 3.51% in the energy recovery efficiency compared with the MPC-CCS, an increase of 9.82% compared with the LQR-CCS, and an increase of 5.38% compared with the MPC-CCS on the bisectional road. From the perspective of energy recovery efficiency, the eMPC-CCS can more effectively invoke the optimized control thresholds in real time by referring to the wheel braking torque corresponding to the optimal road slip ratio, so it has better adaptability to different braking modes. In Table 6 and Table 7, in terms of braking deceleration, the eMPC-CCS has the best performance, which means that the vehicle can slow down to the target speed or stop faster, and allow the driver to control braking effort more accurately.

Through the simulation analysis, the proposed eMPC-CCS can control vehicles to perform anti-lock braking with excellent stability and robustness when the road adhesion coefficient changes suddenly and the road adhesion coefficients on the left and right sides are different.

### 4.3. Hardware-in-the-Loop Test 

To validate the performance of the eMPC-CC and its functionality in lowering computation time and improving real-time performance, a hardware-in-the-loop (HIL) test is conducted. Hardware test planning is depicted in Figure 15, primarily composed of host PC1 (the controller), host PC2 named Speedgoat, and the target machine interface. In host PC1, the eMPC-CCS and vehicle model, including component sub-models, are constructed in MATLAB/Simulink. The Simulink model is then compiled into C code components. Subsequently, the compiled Simulink model from host PC1 is downloaded to PC2 and displayed through the target machine interface. The communication between the controller and host PCs is attained via CAN bus communication. To better illustrate the braking energy recovery performance and real-time performance of the eMPC-CCS, the test scenario involves emergency braking, starting at a speed of 100 km/h and using a braking intensity of 0.8.

Figure 16 represents the slip ratio changes of different CCSs in the test scenario. Table 8 lists the numerical results of the energy recovery efficiency of different CCSs in the test scenario, and the step time costs are shown in Table 9.

Table 8 lists the maximum step time cost, minimum step time cost, and average time cost in the HIL test for different control strategies. Compared with other CCSs, the eMPC-CCS has the smallest step time cost and meets real-time applications in road driving with a maximum computational frequency of more than 50 Hz [49].

In the control strategies, the MPC-CCS, LQR-CCS, and PID-CCS tend to activate the ABS frequently, but the slip ratio curve of the eMPC-CCS is closer to the reference slip ratio. To be specific, from Figure 16, it can be seen that the slip ratio gradually increases during the initial 0.5 s of braking, then vehicle braking torque is adjusted in real-time according to the reference slip ratio. After 3.3 s, the vehicle speed decreases to 10 km/h, at which point the ABS disengages from the braking process, resulting in a sudden surge in the slip ratio to 1. Compared with the slip ratio curve of the other traditional CCSs in the partially enlarged figure, the slip ratio of the eMPC-CCS changes smoothly, so the eMPC-CCS shows a strong adjustment ability. According to the results in Table 9, the eMPC-CCS can increase braking energy recovery efficiency by nearly 3.07% to 9.14% compared with other CCSs.

Through the simulation analyses, the developed eMPC-CCS, formed by offline control laws generation and online control law invocation, can better coordinate the control RBS and ABS, which could follow the braking energy recovering trend of the reference slip ratio. To be specific, in the HIL test, the real-time calculation ability of the eMPC-CCS can also meet the requirement of the communicating frequency for the CAN bus. In short, the eMPC-CCS shows an advantage in coordinated control between RBS and ABS. 

## 5. Discussion

In this study, a novel eMPC-based coordinated control strategy, namely the eMPC-CCS, is proposed for the electro-hydraulic composite braking system. The aim is to ensure the real-time performance and stability of the braking process and maximize the braking energy recovery of the four-wheel-drive hub electric vehicle. Comparative studies are conducted through simulations to verify the feasibility and validity of the eMPC-CCS. From the analysis in Section 3, we can draw the following conclusions:(1)The proposed CCS based on eMPC, named the eMPC-CCS, greatly improves the online calculation speed of coordinated control strategy allocation through offline processing and online implementation, and can provide more accurate and intuitive control performance analysis.(2)A three-dimensional eMPC law generation method based on multiple braking conditions generates a 3D eMPC explicit solution by invoking multiple sets of micro trip segments to generate state feedback control laws, achieving the adaptability of the control strategy.(3)The eMPC-CCS includes an improved eMPC controller with a basic eMPC controller and state error compensator, which improves real-time capability, adaptability, and robustness under various braking conditions. Compared with the other CCSs, namely the PID-CCS, LQR-CCS, and MPC-CCS, the braking energy recovery efficiency of the eMPC-CCS is increased by at least nearly 4%.

## 6. Conclusions

In this paper, a novel eMPC-based coordinated control strategy named the eMPC-CSS is proposed for electro-hydraulic composite braking systems. By combining offline control law generation with online control law invocation, this strategy augments real-time capability and robustness between the RBS and ABS. Offline control law generation, including real-time-oriented state feedback control laws under micro braking segments, supporting the eMPC-CCS to have a properly coordinated control tendency. The online implementation, containing 3D eMPC control law generation and state error compensation, can facilitate control law application in practice while also allowing for the rational distribution of motor and hydraulic braking torque. Compared to other CCSs such as the MPC-CCS, LQR-CCS, and PID-CCS, the proposed eMPC-CSS demonstrates a significant improvement in braking energy recovery efficiency, with gains ranging from approximately 4% to 17%. The simulation-based test and HIL validation verify that the proposed eMPC-CCS effectively ensures the real-time capability, adaptability, and robustness of the CCS, showcasing its anticipated superior performance.

However, it is crucial to acknowledge the significant discoveries and limitations presented in these studies. Firstly, one key limitation identified is the considerable braking torque fluctuations due to the differing dynamic response characteristics of the regenerative and hydraulic braking systems, particularly during mode-switching sequences. This issue warrants further investigation into mode transition-smoothing techniques. Secondly, the current study solely considers the influence of longitudinal force on vehicle braking, disregarding the impact of lateral force throughout the time course. Consequently, it is imperative for future studies to comprehensively examine both the implications of braking torque fluctuations and the influence of lateral force during vehicle braking for a more complete understanding and enhancement of system performance.

## Figures and Tables

**Figure 1 sensors-24-03076-f001:**
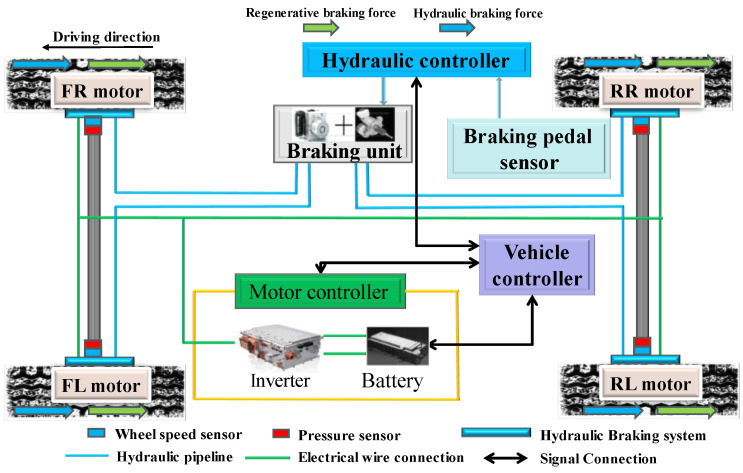
The configuration of a four-wheel hub drive electric vehicle.

**Figure 2 sensors-24-03076-f002:**
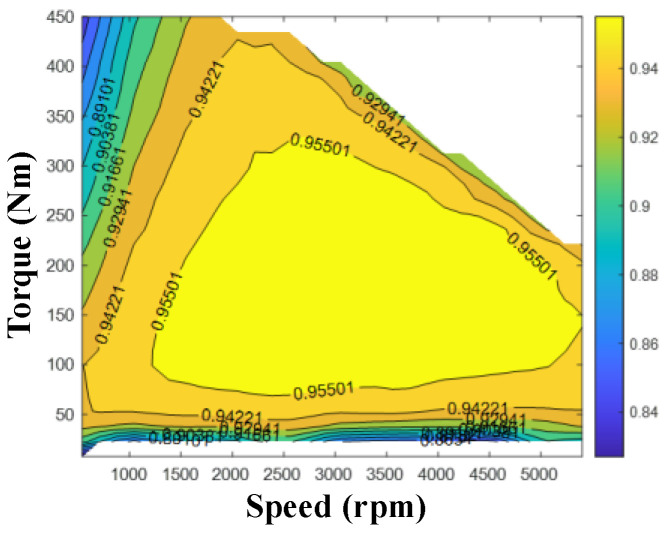
Motor efficiency map.

**Figure 3 sensors-24-03076-f003:**
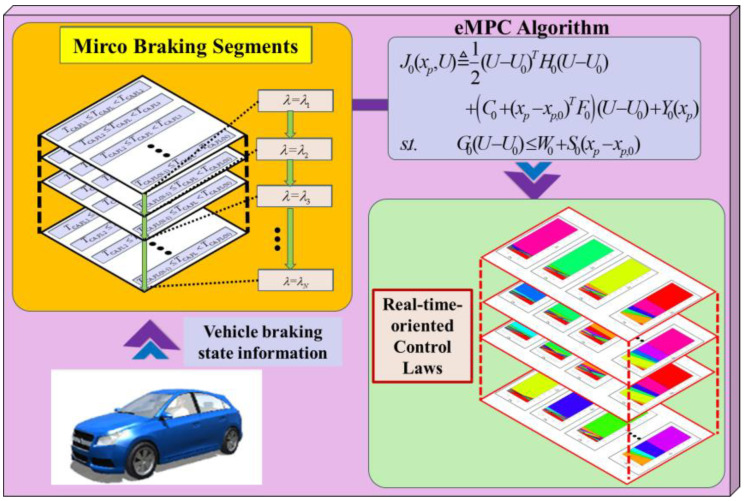
The illustration of the offline control law generation of the eMPC-CCS.

**Figure 4 sensors-24-03076-f004:**
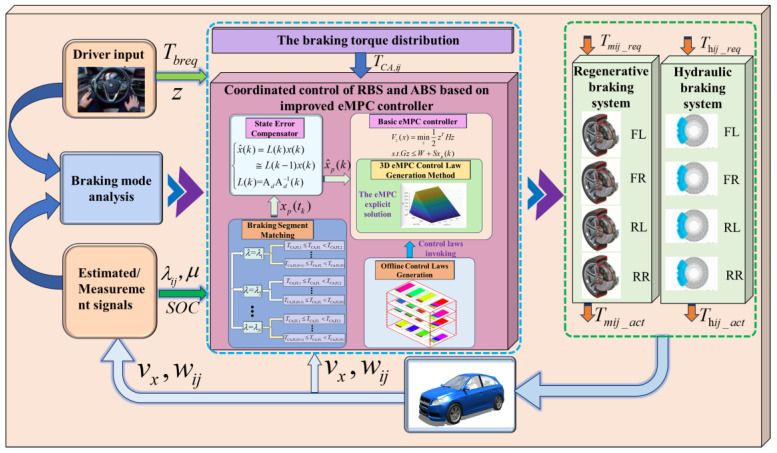
The illustration of the online control law transplantation and state error compensation of the eMPC-CCS.

**Figure 5 sensors-24-03076-f005:**
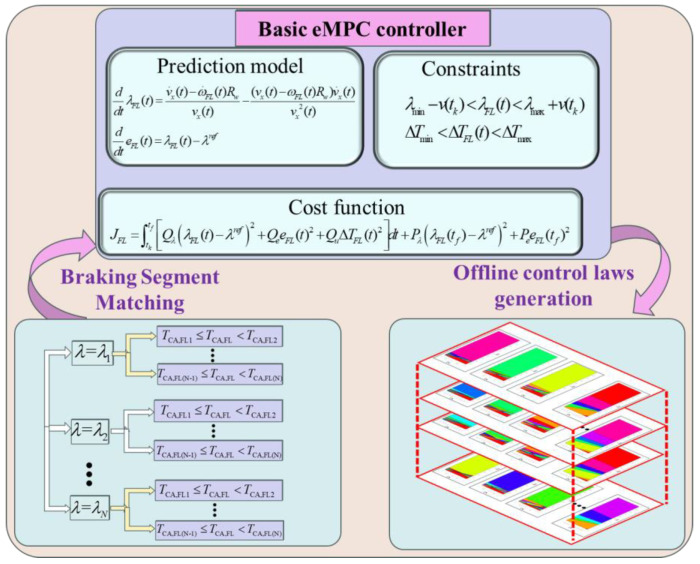
The Generation of state feedback control laws in the basic empc controller.

**Figure 6 sensors-24-03076-f006:**
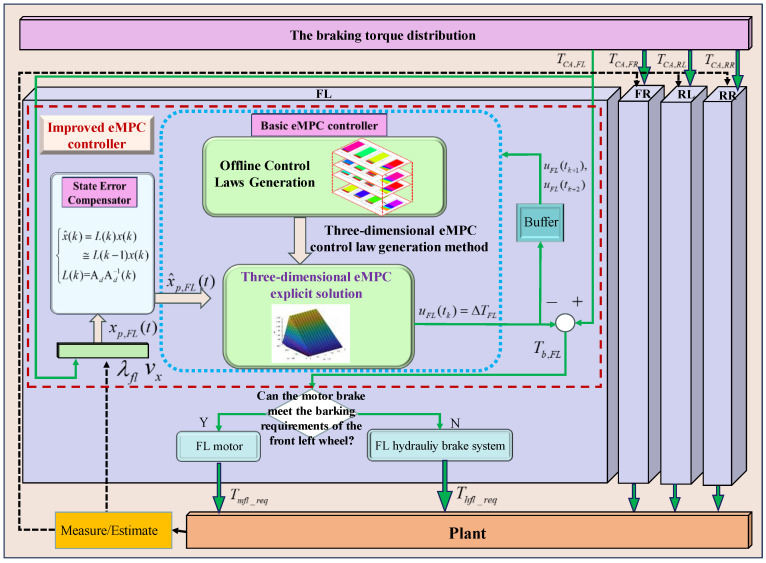
The illustration of the online implementation of the eMPC-CCS.

**Figure 7 sensors-24-03076-f007:**
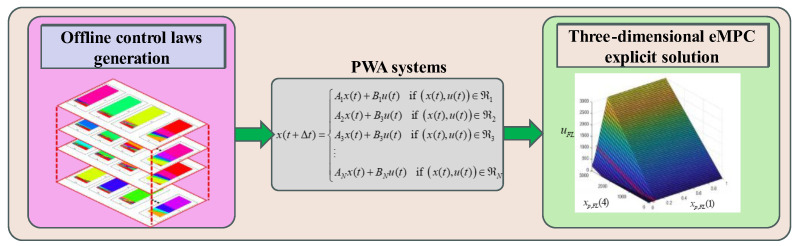
The illustration of the three-dimensional eMPC control law generation method.

**Figure 8 sensors-24-03076-f008:**
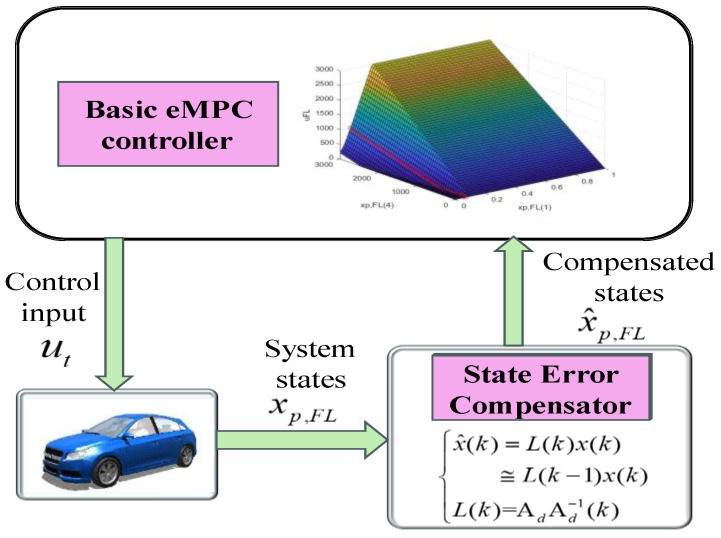
The architecture of the improved eMPC.

**Figure 9 sensors-24-03076-f009:**
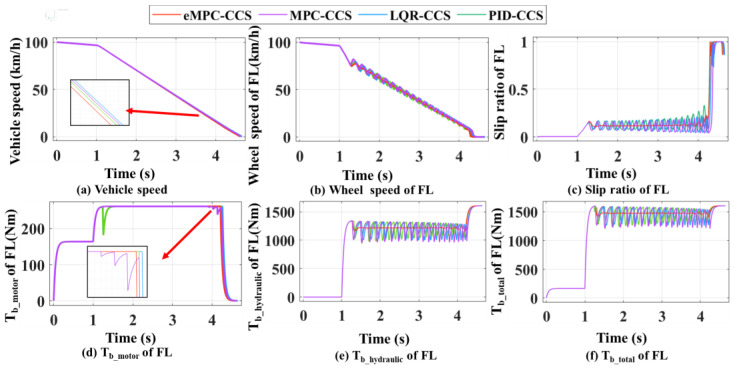
Simulation results of four coordinated control strategies for high-adhesion-coefficient road.

**Figure 10 sensors-24-03076-f010:**
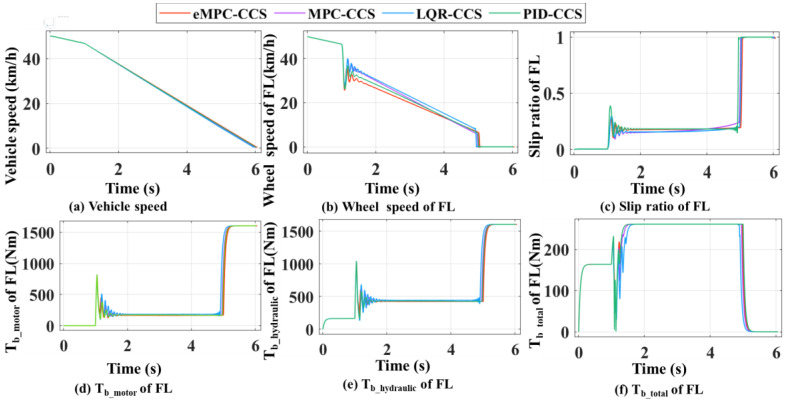
Simulation results of four coordinated control strategies for low-adhesion-coefficient road.

**Figure 11 sensors-24-03076-f011:**
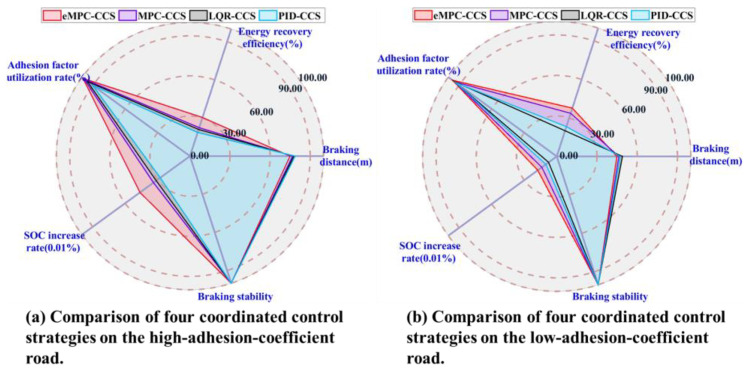
Comparison of four coordinated control strategies on high- and low-adhesion-coefficient roads.

**Figure 12 sensors-24-03076-f012:**
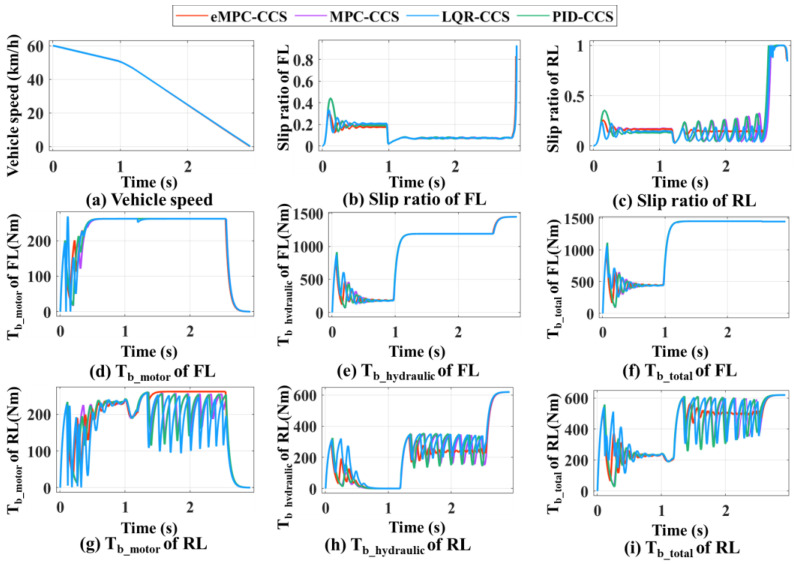
Simulation results of four coordinated control strategies for the joint road.

**Figure 13 sensors-24-03076-f013:**
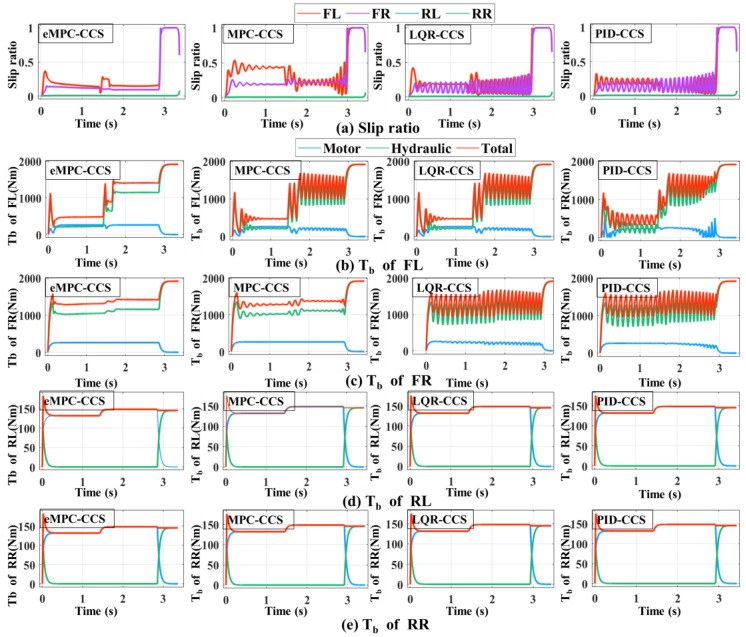
Simulation results of four coordinated control strategies for the bisectional road.

**Figure 14 sensors-24-03076-f014:**
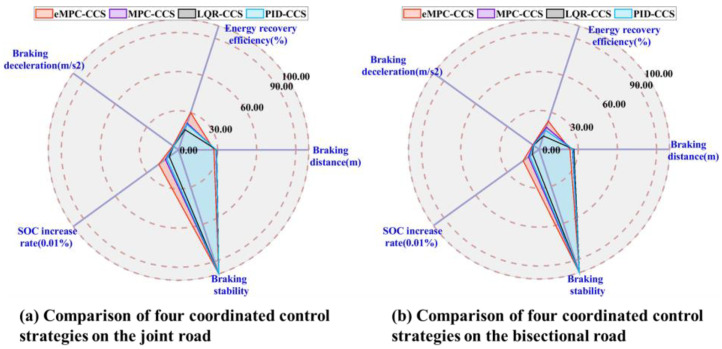
Comparison of four coordinated control strategies on the joint road and the bisectional road.

**Figure 15 sensors-24-03076-f015:**
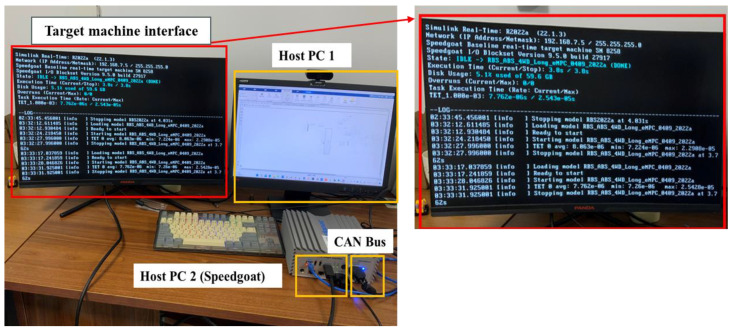
HIL test platform.

**Figure 16 sensors-24-03076-f016:**
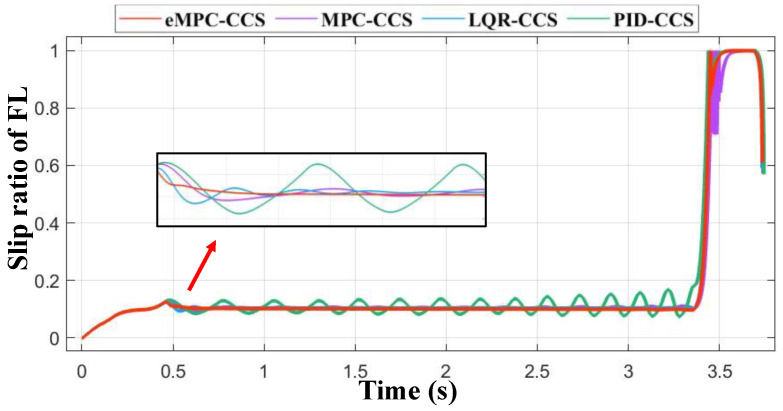
Slip ratio curves by different CCSs.

**Table 1 sensors-24-03076-t001:** Vehicle parameters.

Parameter	Sign	Value
Vehicle mass	m	2508 [kg]
Wheel radius	$R_{w}$	0.377 [m]
The horizontal distance from the center of gravity to the front axle	$l_{f}$	1.506 [m]
The horizontal distance from the center of gravity to the rear axle	$l_{r}$	1.483 [m]
Wheelbase	l	2.992 [m]
Center of gravity height	$h_{g}$	0.58 [m]
Wheel moment of inertia	$J_{w}$	4299 [kgm^2^]

**Table 2 sensors-24-03076-t002:** Intermodal parameters.

Symbol	Parameters	Value
$m_{i}$	Apparent front (rear) corner mass (kg)	625 (629)
$F_{z,i}$	Front (rear) tire vertical load (N)	6125 (6164)
B	MF coefficient: stiffness factor	21.52
C	MF coefficient: shape factor	1.4
D	MF coefficient: peak value	0.45
E	MF coefficient: curvature factor	−0.28
$N_{p}$	Prediction step	3
$t_{r}$	Sampling time (s)	0.004

**Table 3 sensors-24-03076-t003:** Implementation structure of the error compensation.

Initialization:
From the system Equation (35), set $A_{d}[0]=A_{d}$$,\;B_{d}[0]=B_{d}$. For the system Equation (41) obtain the optimal look-up map, i.e., the eMPC solution in Figure 7 by solving the mp-QP problem Equation (10). Set the initial condition x(0) of the system Equation (35) and k=0. Step 1: Measure the state vector x(k) of the system Equation (35). Step 2: If k=0 then $A_{d}(-1)=A_{d}$. Otherwise, $A_{d}(-1)$ is calculated by using Equation (35) with x(k), x(k−1), u(k−1) if ranks of $D_{x}(k-1)$ and $\left[D_{x}(k-1),x(k)-A_{d}x(k-1)-B_{d}x(k-1)\right]$ are equal for all k≥1. Step 3: Obtain L(k−1) by substituting $A_{d}[k-1]$ from Equation (44) for $A_{d}[k]$ in Equation (46), and estimate $\hat{x}(k)$ by Equation (47) with x(k) and L(k−1). Note that, by Equation (46), $L(-1)=A_{d}^{-1}A_{d}(-1)=A_{d}^{-1}A_{d}=I$. Step 4: Evaluate the function u(k) by eMPC solution in Figure 7 based on $\hat{x}(k)$ and apply u(k) to the system Equation (35). Step 5: Set k=k+1, and go to Step 1.

**Table 4 sensors-24-03076-t004:** Comparison of four coordinated control strategies on high-adhesion-coefficient road.

Control Strategy	Energy Recovery Efficiency (%)	Braking Distance (m)	Adhesion Factor Utilization Rate (%)	Final SOC of the Battery (%)	Braking Stability (%)
eMPC-CCS	30.38	75.94	98	80.37	100
MPC-CCS	22.36	76.88	96.6	80.27	100
LQR-CCS	20.99	77.79	95	80.25	100
PID-CCS	18.63	78.79	93	80.22	100

**Table 5 sensors-24-03076-t005:** Comparison of four coordinated control strategies on low-adhesion coefficient road.

Control Strategy	Energy Recovery Efficiency (%)	Braking Distance (m)	Adhesion Factor Utilization Rate (%)	Final SOC of the Battery (%)	Braking Stability (%)
eMPC-CCS	37.64	45.83	96	80.14	100
MPC-CCS	33.59	46.29	94.8	80.11	100
LQR-CCS	20.02	48.98	94.1	80.06	100
PID-CCS	25.21	47.32	94.5	80.08	100

**Table 6 sensors-24-03076-t006:** Comparison of four coordinated control strategies on the joint road.

Control Strategy	Energy Recovery Efficiency (%)	Braking Distance (m)	Braking Deceleration (m/s^2^)	Final SOC of the Battery (%)	Braking Stability (%)
eMPC-CCS	23.68	24.08	5.89	80.12	100
MPC-CCS	18.05	25.68	5.5	80.08	100
LQR-CCS	10.82	26.7	5.2	80.05	100
PID-CCS	15.35	26.2	5.34	80.07	100

**Table 7 sensors-24-03076-t007:** Comparison of four coordinated control strategies on the bisectional road.

Control Strategy	Energy Recovery Efficiency (%)	Braking Distance (m)	Braking Deceleration (m/s^2^)	Final SOC of the Battery (%)	Braking Stability (%)
eMPC-CCS	26.07	27.18	5.11	80.13	100
MPC-CCS	22.56	28.9	4.81	80.10	100
LQR-CCS	16.25	29.31	4.74	80.06	100
PID-CCS	20.69	29.16	4.76	80.09	100

**Table 8 sensors-24-03076-t008:** Braking energy recovery efficiency of different CCSs.

Control Strategy	Energy Recovery Efficiency (%)	Braking Distance (m)	Final SOC of the Battery (%)	Braking Stability (%)
eMPC-CCS	25.58	47.18	80.24	100
MPC-CCS	22.51	48.9	80.20	100
LQR-CCS	20.61	49.31	80.18	100
PID-CCS	16.44	52.16	80.15	100

**Table 9 sensors-24-03076-t009:** Real-time applications of different CCSs.

Control Strategy	Minimum Step Time Cost (s)	Maximum Step Time Cost (s)	Average Step Time Cost (s)
eMPC-CCS	7.26 × 10^−6^	2.54 × 10^−5^	7.76 × 10^−6^
MPC-CCS	7.5 × 10^−6^	2.73 × 10^−5^	8.27 × 10^−6^
LQR-CCS	7.25 × 10^−6^	3.02 × 10^−6^	7.78 × 10^−6^
PID-CCS	7.33 × 10^−6^	2.71 × 10^−5^	7.91 × 10^−6^

## Data Availability

No new data were created or analyzed in this study. Data sharing is not applicable to this article.

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
