# Peer review of "RBS and ABS Coordinated Control Strategy Based on Explicit Model Predictive Control"

_sensors, 2024, doi:10.3390/s24103076_

Round 1
Reviewer 1 Report
Comments and Suggestions for Authors
The manuscript presents an interesting coordinated control strategy (CCS) based on explicit model predictive control (eMPC), aiming to enhance the coordinated control effectiveness of the regenerative braking system (RBS) and the anti-lock braking system (ABS) of an electro-hydraulic composite braking system.
The topic is interesting, and the quality of the current version of the paper is sufficiently good. The authors state clearly their contributions among which I believe an important one is the approach used to validate the performance of their proposed eMPC-based strategy, based on a hardware-in-the-loop (HIL) test.
The authors have also established a MATLAB/Simulink simulation model for a four-wheel hub drive electric vehicle.
The experimental results demonstrate sufficiently the efficiency of their proposed approach in terms of braking energy recovery and real-time capability.
The manuscript is clear, relevant for the field and presented in a well-structured manner. Related work is sufficiently investigated. The discussion addresses sufficiently the main research question posed and is consistent with the contributions presented. A large part of the cited references are mostly recent publications and relevant to the subject.
Just a few comments.
0n lines 260, 263, 294, 306, 339, 373, 374, 377, 378, 392, 395, 410, 411, 415, 421, 423, 424, 426, 427, 428, 444, 448 and particularly on page 17 there are several references errors in equations numbers (… Eqn. Error! Reference source not found. ).
On line 555 the equation number (1) does not seem to be correct.
On line 619 correct “As shown in Fig. 14, Compared” to “As shown in Fig. 14, compared”.
The text in Acknowledgments needs to be taken care or removed.
Comments on the Quality of English LanguageEnglish language is fine.
Reviewer 2 Report
Comments and Suggestions for Authors
This manuscript paid attention to a new coordinated control strategy (eMPC-CSS) based on eMPC for electro-hydraulic composite braking systems. By combining offline control law generation with online control law invocation, this strategy enhances the real-time capability and robustness between RBS and ABS. Offline control law generation, including real-time oriented state feedback control law under the micro-braking section, supports eMPC-CCS to generate appropriately coordinated control trends. This work is interesting. If possible, the following issues should be stressed properly:
1. In Section 2.4, the author mentions that the effect of temperature on the operation of the battery is ignored. It is necessary to briefly explain why these simplifications were made.
2. In the first paragraph at the beginning of Section 3.1, it is mentioned that the eMPC algorithm is used for offline calculations. In this paragraph, eMPC should be briefly introduced so that readers can more clearly understand the implementation principle of eMPC.
3. In Fig. 4, the part of the braking segment matching is not clearly. It is recommended to make the necessary modifications to improve clarity.
4. In Section 3.2.1, the explicit solution of RBS and ABS coordinated control can be described in more detail, especially regarding how the offline generated control laws are invoked through the Piecewise Affine (PWA) systems.
5. In Table 3, the font size and spacing between each line are not consistent. It is recommended to make the necessary modifications.
6. Concerning the fast MPC, some references should be considered, such as
Chu H, Meng D, Huang S, et al. Autonomous high-speed overtaking of intelligent chassis using fast iterative model predictive control[J]. IEEE Transactions on Transportation Electrification, 2023.
Meng D, Chu H, Tian M, et al. Real-Time High-Precision Nonlinear Tracking Control of Autonomous Vehicles Using Fast Iterative Model Predictive Control[J]. IEEE Transactions on Intelligent Vehicles, 2024.
Comments on the Quality of English LanguageThe english is fine.
Round 2
Reviewer 1 Report
Comments and Suggestions for Authors
No further comments.
Reviewer 2 Report
Comments and Suggestions for Authors
My concerns are well addressed.